# Pointwise error estimates in localization microscopy

Martin Lindén[1],[*], Vladimir Ćurić[1],[*], Elias Amselem[1] & Johan Elf[1]

Pointwise localization of individual fluorophores is a critical step in super-resolution localization microscopy and single particle tracking. Although the methods are limited by the localization errors of individual fluorophores, the pointwise localization precision has so far been estimated using theoretical best case approximations that disregard, for example, motion blur, defocus effects and variations in fluorescence intensity. Here, we show that pointwise localization precision can be accurately estimated directly from imaging data using the Bayesian posterior density constrained by simple microscope properties. We further demonstrate that the estimated localization precision can be used to improve downstream quantitative analysis, such as estimation of diffusion constants and detection of changes in molecular motion patterns. Finally, the quality of actual point localizations in live cell super-resolution microscopy can be improved beyond the information theoretic lower bound for localization errors in individual images, by modelling the movement of fluorophores and accounting for their pointwise localization uncertainty.

[1] Department of Cell and Molecular Biology, Uppsala University, Box 596, 751 24 Uppsala, Sweden. * These authors contributed equally to this work. Correspondence and requests for materials should be addressed to J.E. (email: johan.elf@icm.uu.se).

Super-resolution fluorescence microscopy and live cell single particle tracking (SPT) rely on computer intensive data analysis to find and localize single fluorescent emitters in noisy images. Much effort has been spent on developing and testing efficient spot localization algorithms[1] and understanding the theoretical limits for localization accuracy[2–5]. However, the problem of estimating and using the actual precision is still unsolved.

PALM/STORM-type super-resolution imaging[6,7] relies on the serial activation and localization of sparse photo-switchable fluorophores. Knowledge about the localization precision is important to build up a high resolution image since uncertain localizations will only contribute blur. Often, only the number of photons, pixel size and background noise for each emitter is used to estimate the precision, assuming that it achieves its theoretical limit. However, theoretical estimates neglect many important factors, and are prone to systematic errors in particular when the background is variable and the emitter is moving, which is the common situation for live cell super-resolution imaging.

Knowledge of the localization precision is also important in SPT[8], where it can be used to improve estimators of diffusion constants[9,10]. It is common in live cell imaging that the spot quality varies throughout an experiment, for example, due to out-of-focus motion, drift, motion blur, fluorophore intensity fluctuations, heterogeneous background or gradual photo-bleaching of the background or labelled molecule.

Here, we investigate methods to extract and use localization precision of single spots in super-resolved SPT, using a combination of experimental data and highly realistic simulated microscopy experiments[11]. We characterize precision estimators based on Gaussian spot models, and find that a Bayesian approach that incorporates basic information about physical limitations in the detection system outperforms estimators based on maximum-likelihood localizations and the Cramér-Rao lower bound (CRLB)[4,5]. We then demonstrate how precision estimates can be used to improve parameter inference, event detection, and localization errors in SPT data, and give a variational expectation maximization (EM) algorithm for a diffusive hidden Markov model (HMM) which extends previously described algorithms[9,10,12–16] by accounting for multi-state diffusion, localization uncertainty and motion blur.

## Results

**Pointwise precision with maximum-likelihood estimates.** Estimating localization precision is closely related to estimating positions, where the maximum-likelihood estimate (MLE) is generally considered the optimal method. A maximum-likelihood method starts with a likelihood function, that is, the probability density function of a probabilistic model for generating images of spots (pixel counts in a small region around a spot) with the emitter position among the adjustable parameters. The MLE is the set of parameters that maximize the likelihood function for a particular spot image. Following common practice, we model electron multiplying CCD (EMCCD) camera noise with the high-gain approximation plus Gaussian readout noise[4,17] (see Methods). The spot shape is modelled by a symmetric Gaussian intensity profile plus a constant background intensity. The fit parameters are thus spot position ($\mu_x$, $\mu_y$), background intensity $b$, spot width $\sigma$ and spot amplitude $N$ (see Methods, equation (2)), while the camera noise parameters are assumed known from camera calibration.

The localization error is the difference $\mu_{est.} - \mu_{true}$ between estimated and true positions. The precision describes the statistical distribution of the error, either in a Bayesian posterior sense, or in the frequentist sense of repeated localizations of equivalent spots. The precision is related to the shape of the

likelihood maximum: a sharply peaked maximum means that only a narrow set of parameters are likely, while a more flat maximum means greater uncertainty and lower precision.

The CRLB is the smallest possible variance of an unbiased estimator for a given set of model parameters, and is related to the expected sharpness of the likelihood maximum. While this is strictly speaking not a statement about a single image, but rather about the average information content of data generated by a model, it is often used to estimate localization precision. We use an accurate analytical approximation developed by Rieger and Stallinga[5] (see Methods, equation (5)). A Bayesian alternative to the CRLB is to consider the posterior distribution of the fit parameters for a particular image. We use the Laplace approximation[18] to derive an approximate Gaussian posterior from the likelihood maximum (see Methods, equation (6)). Both estimators estimate the root mean square error (RMSE), but none of them are well characterized as estimators of localization precision.

To test these estimators, we analysed simulated movies of a fluorescent particle diffusing at $D = 1\,\mu m^2\,s^{-1}$ in an *E. coli*-like geometry. The movies cover a broad range of experimentally relevant imaging conditions and include realistic EMCCD noise, background fluorescence, a non-Gaussian vectorial-based point-spread function (PSF)[19,20] for isotropic or rotationally mobile emitters[21] (see Methods). Examples of simulated spots are shown in Fig. 1.

Motion blur effects depends on the relative strengths of several parameters. A simple scaling argument to gauge its importance is to ignore out-of-plane motion and compare the s.d. of a Gaussian diffraction-limited spot to that of the fluorophore's in-plane diffusion path during the exposure time $t_E$. The latter is given by $\sqrt{Dt_E/3}$ (ref. 22), and the spot width $\sigma_0$ is related to the wavelength $\lambda$ and numerical aperture (NA) through[23]

$$\sigma_0 \approx 0.21\lambda/NA. \tag{1}$$

We consider $\sqrt{Dt_E/3\sigma_0^2} < 0.5$ to be weak blur, which corresponds to $t_E < 6\,ms$ in our case.

A basic consistency check for any precision estimator is that the estimated and true RMSE agree. Such a comparison is shown in Fig. 2a for MLE localizations in a range imaging conditions

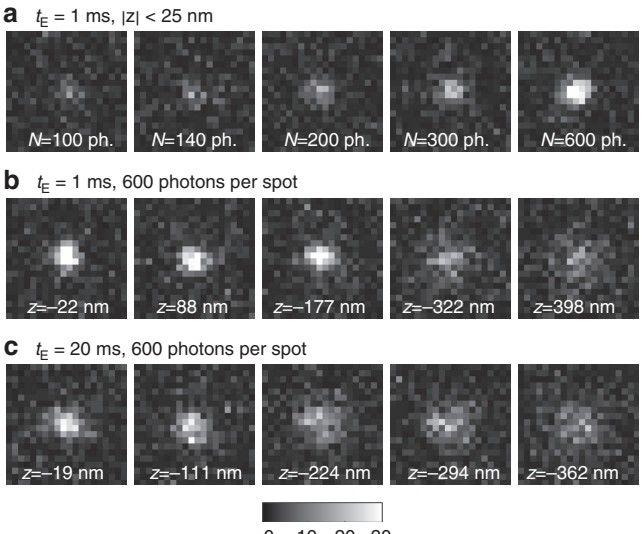

**a** $t_E = 1\,ms$, $|z| < 25\,nm$

N=100 ph.　N=140 ph.　N=200 ph.　N=300 ph.　N=600 ph.

**b** $t_E = 1\,ms$, 600 photons per spot

z=−22 nm　z=88 nm　z=−177 nm　z=−322 nm　z=398 nm

**c** $t_E = 20\,ms$, 600 photons per spot

z=−19 nm　z=−111 nm　z=−224 nm　z=−294 nm　z=−362 nm

0　10　20　30

**Figure 1 | Factors that influence localization precision.** Simulated images of a diffusing fluorophore with diffusion constant $1\,\mu m^2\,s^{-1}$, 1 photon per pixel background, EMCCD noise and varying localization uncertainty due to varying (**a**) spot amplitude $N$, (**b,c**) average defocus $z$ and exposure time $t_E$. Image size 20-by-20 pixels, pixel size 80 nm, colorbar indicates photons per pixel.

with short exposure times. Both estimators show <10% discrepancy under good conditions, where the spots are bright and the average errors low. However, the CRLB formula deteriorates significantly as conditions worsen and the errors increase, because of the worse performance on defocused spots (Supplementary Fig. 1).

A stricter criterion is conditional consistency, meaning that estimated and true RMSE are consistent for each value of the estimated precision. Such a comparison is shown in Fig. 2b. Here, the box plot shows the range of results for different imaging conditions. We recognize the deterioration of the CRLB at high estimated RMSE. In addition, the wide boxes at low RMSE show a bias towards underestimating the precision in some conditions.

**Pointwise precision with maximum a posteriori estimates**. Can the MLE precision estimates be improved? One clue is that the distribution of spot widths from MLE fits (Fig. 2c) contains a sizable fraction of spots more narrow than the width $\sigma_0$ of a diffraction-limited spot, which is unphysical. Indeed, the Laplace estimator performs better on the sub-population of fits with $\sigma \geq \sigma_0$ (Supplementary Note 1). However, using this criterion to

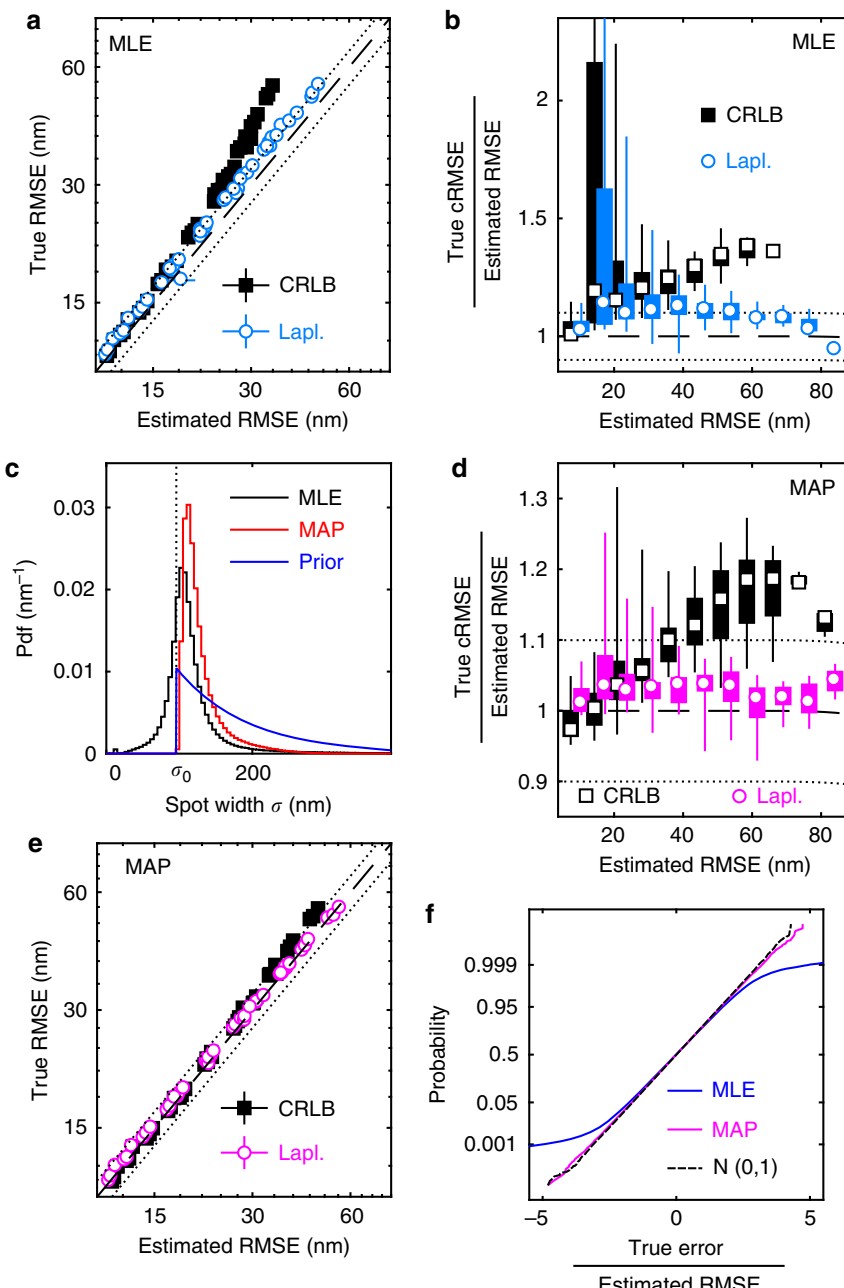

**Figure 2 | True and estimated precision at short exposure times.** (**a**) True and estimated RMSEs for MLE localizations in different imaging conditions. Precisions are estimated using the CRLB and Laplace estimators, and error bars (mostly smaller than the symbols) indicate bootstrap s.e.m. Dashed and dotted lines indicate 0% and ±10% bias, respectively. (**b**) Conditional RMSE (cRMSE) normalized by estimated RMSE. Symbols, boxes and lines show the median, 25% and 75% quantiles, and the whole span for all localization conditions. (**c**) Distribution (probability density function, pdf) of fitted spot widths $\sigma$ for the MLE and MAP fits, and spot width prior. The theoretical minimum width $\sigma_0$ is indicated by a vertical line. (**d,e**) Same as in **a,b**, but for MAP localizations. (**f**) Probability plots of localization errors normalized by Laplace RMSE estimates for MLE and MAP fits. Only spots with estimated RMSE smaller than 3 pixels are included, and a standard normal distribution, $N(0,1)$, is included as reference.

exclude points from the analysis is not a practical solution, as too few points will remain. To utilize this physical insight while retaining more spots, we instead constrain the parameter fits using prior distributions, thus replacing MLE with maximum a posteriori estimation (MAP). This allows some parameter fluctuations to model intrinsic variations in the size and shape of spots. To study only the effects of fluorophore motion and varying imaging conditions, we run numerical experiments where we use high photon counts to suppress purely statistical fluctuations. The results (Supplementary Note 2) indeed show the spot widths confined to a finite interval above $\sigma_0$. However, the full distribution of all fit parameters is complex and strongly dependent on experimental conditions. A prior modelled on this basis would therefore be difficult to construct, and only applicable in a narrow range of conditions.

Given the complications inherent in using a highly detailed prior based on, for example, high intensity simulations, we instead seek a less informative and more general prior to regularize the problem, incorporate relevant scale information and exclude unreasonable parameter values. We reparameterize the model to enforce the lower bound on the spot width in each fit, set an exponential prior on the dimensionless excess spot width $(\sigma - \sigma_0)/\sigma_0$, and a weak log-normal prior on the background. User input is limited to easily accessible quantities: $\sigma_0$ (via equation (1)), and an order-of-magnitude background estimate (Supplementary Note 3) where simple inspection or a local estimator[24] should suffice. The resulting MAP estimator performs better in all aspects, as seen in Fig. 2c–e.

So far, we have looked at averages, but the full distribution of errors is also interesting. In particular, most[9,10,12–16] (but not all[25]) recent statistical models of SPT data assume Gaussian errors, although this assumption has not been tested. If the Laplace approximation (equation (6)) was exact, the errors normalized by the estimated RMSE would be Gaussian with unit variance, and produce a straight line in the Gaussian probability plot in Fig. 2f. The MLE results only agree partly with the reference unit normal, consistent with a sub-population of fits with underestimated precision, but the MAP results show good agreement.

We also generalized the approach to an asymmetric Gaussian spot model (see Methods), which performs better than the symmetric one with increased motion blur, as seen in Fig. 3. Both priors also improve localization errors and convergence rates compared to the MLE fits (Supplementary Note 4). However, the asymmetric prior is less robust w.r.t. its parameterization (Supplementary Note 3). The combination of long exposure and high spot intensity also remains difficult in either case, but this can often be avoided experimentally, for example by decreasing the exposure time.

A possible further development is to improve the Gaussian localization model, perhaps using an experimentally derived PSF model[26]. To explore this, we experimented with data using a Gaussian spot model for both PSF simulation and localization (Supplementary Note 5). We see only modest improvements, however, and as experimentally derived PSFs are instrument-specific, we do not pursue this further.

Overall, these results show that pointwise precision estimates using the Laplace approximation works well in a wide range of experimentally relevant conditions, if aided by some basic information about PSF shape and background. In this case, we see good support for the assumption of Gaussian-distributed localization errors.

**Validation on real data**. To test the above conclusions on real data, we imaged immobilized fluorescent beads, alternating strong and weak excitation as shown in Fig. 4a. We used images under strong excitation conditions to extract an approximate ground truth for testing the precision estimates in the dim images. We estimated the position and precision of spots using the estimation procedures described above with a symmetric spot model, except for changing the background prior to be centred around the mean background (0.7 photons per pixel) seen in dim frames. A drift-corrected ground truth was estimated by linear interpolation between the mean positions obtained from each block of 10 consecutive bright images. In addition, the intensity differs by about a factor 10 between bright and dim frames. Overall, the RMSEs of the ground truth should therefore be approximately 10-fold lower than that of a single dim spot.

Figure 4e shows the resulting comparison between true and estimated precision, with every point corresponding to a single bead. It qualitatively reproduces the behaviour on simulated images in Fig. 2, confirming our conclusion that the Laplace approximation is preferable to the CRLB formula as a precision estimator, and that the good performance of our prior is not limited to that particular set of simulated data.

**Estimating diffusion constants**. Next, we consider how precision estimates can improve estimates of diffusion constants, arguably the most common analysis of SPT data. Using simulated data, we estimated positions and precisions using the asymmetric MAP-Laplace estimators described above, extracted uninterrupted trajectories with ten steps, and finally estimated diffusion constants using the covariance-based estimators of ref. 9 (see Methods) with and without the use of precision estimates. Figure 5 shows the resulting mean value and 1% quantiles under varying imaging conditions, plotted against the signal-to-noise ratio, which is defined as half the diffusive step-length s.d. divided by the RMSE[9]. The use of estimated precision obviously improves the variability of the diffusion estimates substantially. The covariance-based estimators only use the average precision in each trajectory. We also implemented a maximum-likelihood estimator for the diffusion constant[10] that makes explicit use of pointwise precisions (see Methods), but found no further improvement (Supplementary Fig. 2).

**Analysis of multi-state data**. We now turn to a more challenging problem where pointwise precision does matter: data where both the diffusion constant and localization error change significantly on similar time scales. In SPT, changes in diffusion constant can be used as a non-invasive reporter on intracellular binding and unbinding events[27]. However, diffusive motion and localization errors contribute additively to the observed step-length statistics (equation (7)), and thus changes in diffusion constants and localization errors cannot be reliable distinguished.

As an example, we consider a protein that alternates between free diffusion $(D = 1 \, \mu m^2 \, s^{-1})$ and a bound state simulated by slow diffusion $(D = 0.1 \, \mu m^2 \, s^{-1})$. We study an ensemble of trajectories with four binding/unbinding events, two of which occur about 400 nm out of focus, and thus are accompanied by substantial broadening of the PSF and increases in localization errors. This defocus matches roughly the radius of an *E. coli* cell, and the scenario could model tracking experiments with cytoplasmic proteins that can bind to the inner cell membrane.

Using SMeagol[11], we simulated 10,000 replicates of the above set of events, at a camera frame rate of 200 Hz, continuous illumination, and 300 photons per spot on average. Figure 6a shows the $z$ coordinates in the input trajectory, and the framewise RMSE produced by asymmetric MAP localization as described above. Different replicates contain identical reaction events, but differ in the microscopic diffusion paths as well as noise

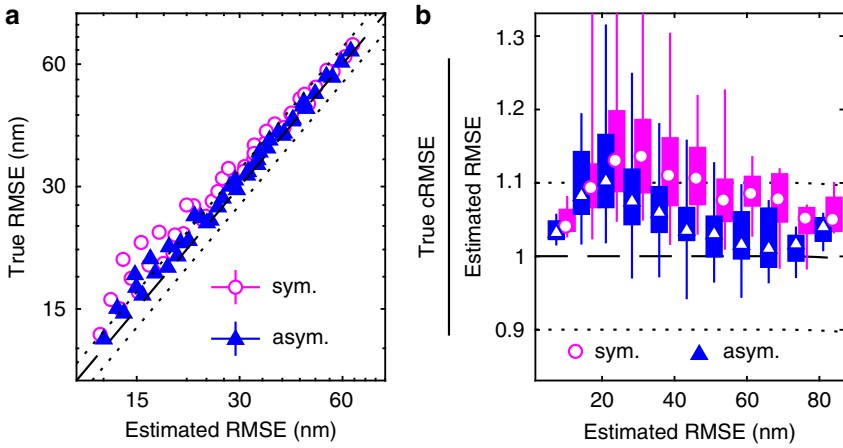

**Figure 3 | True and estimated precision at longer exposure times.** Results for $0.6 < \sqrt{Dt_E/3\sigma_0^2} < 0.9$, using symmetric and asymmetric Gaussian spot models for MAP localization and Laplace precision estimates. Dashed and dotted lines indicate 0% and ±10% bias, respectively. (**a**) True and estimated RMSEs for each imaging condition with error bars (mostly smaller than symbols) showing bootstrap s.e.m. (**b**) Conditional RMSE (cRMSE) normalized by estimated RMSE. Symbols, boxes, and lines show the median, 25% and 75% quantiles, and the whole span for all localization conditions.

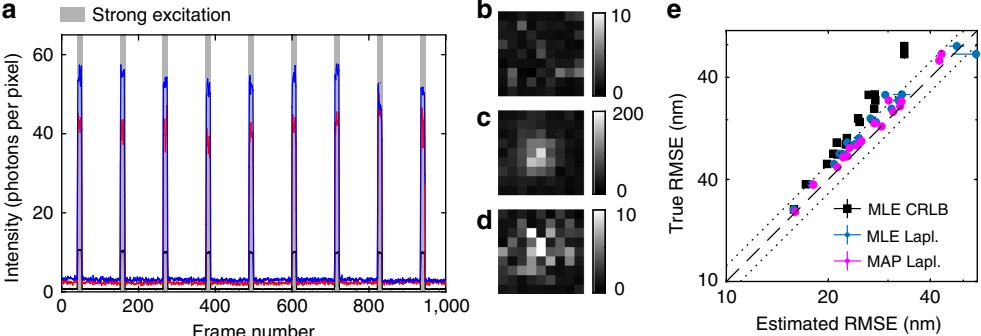

**Figure 4 | Validating estimators of localization precision using real data.** (**a**) Intensity in different frames for two different beads (red, blue) and the background (black). Grey areas indicate periods of strong excitation intensity. (**b**–**d**) Image examples, 9-by-9 pixels, 80 nm per pixel, colorbar indicates photons per pixel. (**b**) Background in high intensity frame, and a bead in a (**c**) high intensity and (**d**) a low intensity frame. (**e**) True and estimated RMSEs for individual beads, using MLE localization and CRLB precision estimates, MLE localization and Laplace precision estimates, and MAP localization and Laplace precision estimates. Error bars indicate bootstrap s.e.m. (mostly smaller than the symbols).

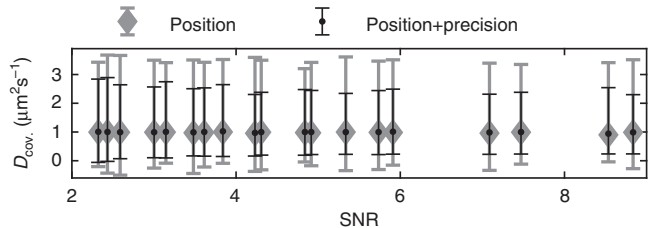

**Figure 5 | Estimating diffusion constants with and without precision estimates.** Mean value and 1% quantiles of covariance-based diffusion constant estimates $D_{cov.}$ from simulated 10-step trajectories plotted against the signal-to-noise ratio (SNR). The true value is $1\,\mu m^2\,s^{-1}$.

realizations. Examples of simulated spots along a trajectory are shown in Fig. 6b.

To analyse this challenging data set, we extend the Berglund model for diffusing particles[28] to multiple diffusion states governed by a HMM, for which we derived a variational EM algorithm (see Methods). We then analysed each simulated trajectory with three different two-state HMMs: first, the extended Berglund model, which explicitly models motion blur and pointwise localization errors. Second, a Kalman-type limit of the Berglund model, which models pointwise localization errors

but not blur effects. This is an interesting comparison, since multi-state Kalman-type algorithms have been studied previously[13,15,16]. Third, the variational Bayes single particle tracking software (vbSPT), which neglects both blur effects and localization errors[27]. We do not consider the ability of vbSPT to estimate the number of diffusive states. Figure 6c shows the inferred average state from the three different methods. As expected, the two HMMs that include localization errors outperform vbSPT at detecting the strongly defocused first and third binding events. The Berglund model does not give the best classification of the two short binding events. However, it does give the lowest overall misclassification rate, 9.3% versus 10.1% and 19% for the Kalman and vbSPT models, respectively.

Next, we look at estimated diffusion constants. Here, the Kalman and vbSPT models make systematic errors as seen in the bare parameters in Fig. 6d. However, by comparing the step-length statistics between the different models, one can derive heuristic correction factors (see Methods, equation (9)) which reduce the bias substantially, as shown in Fig. 6e.

To finally compare the different HMMs on more well-behaved data, we reran the same experiment but with all $z$ coordinates rescaled by a factor 1/5 in the PSF model (Fig. 6f,g), which removes most of the $z$-dependent defocus effects. On this less challenging data set, event detection is much improved and the

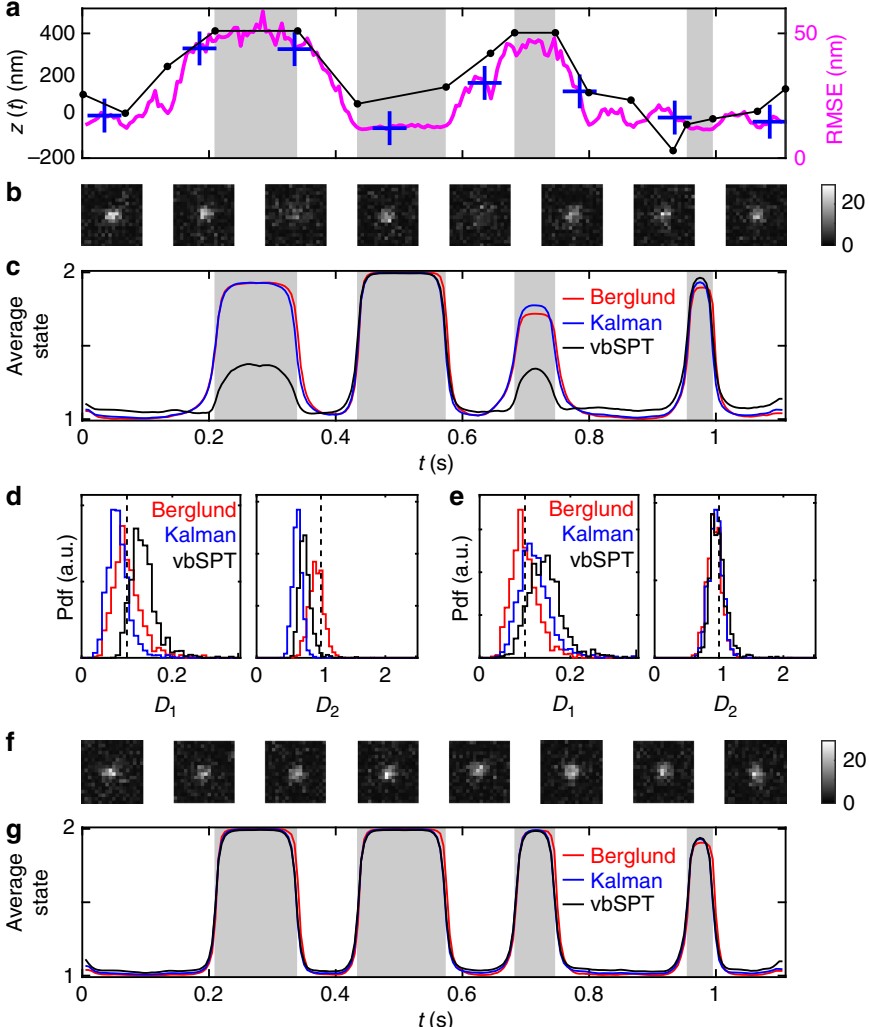

**Figure 6 | Detecting binding events in and out of focus. (a)** Input $z$ coordinates and RMSEs, and **(b)** representative spot images along simulated trajectories, from time points indicated by $+$ in **a**. Grey areas indicate binding events. **(c)** Average state occupancy for the different HMMs. **(d)** Distribution of estimated diffusion constants for the two states (probability density function, pdf), with true values indicated by dashed lines. **(e)** Corrected diffusion constant estimates. **(f)** Spot images (rendered as in **a**) and **(g)** HMM occupancies for trajectories simulated without defocus effects. Spot images are 18-by-18 pixels, with 80 nm pixel size, and the colour bar indicating number of photons per pixel.

differences between the three HMMs are much less pronounced, although the Berglund model still has a slight edge in overall misclassification (4.7% versus 6.5%, and 7.8%).

We conclude that explicit use of pointwise localization errors make a significant improvement if these errors vary a lot in the data, while the more accurate description of blur effects in the Berglund model is a more incremental improvement.

**Position refinement**. Since the new HMM includes the true trajectory as a hidden variable and performs a global analysis, it can be used to refine individual localized positions, and in principle beat the CRLB for single-image localizations. In essence, if a particular position is estimated to be highly uncertain, and the molecule is moving slowly, it may be better to estimate its position using the average localizations in neighbouring frames. Figure 7a illustrates the true, measured and refined positions for part of a two-state trajectory. Figure 7b shows the relative change of the RMSE for each frame in Fig. 6a after refinement, and includes improvements of up to 50%. Large localization errors and small diffusion constant lead to larger relative improvement, as expected.

## Discussion

Fluorophore positions are not the only useful kind of information in super-resolution microscopy images. Here, we have shown that pointwise localization precision can also be extracted and used to improve quantitative data analysis. This is particularly important for live cell data, where molecules and structures are moving, and constraints on labelling and imaging often mean less bright spots compared to fixed and stained cells.

In general, our results show that estimating localization precision is harder than the localization problem itself, but still feasible. The performance may in fact be somewhat improved in real applications, since spot detection algorithms tend to discard the least well-behaved spots (Supplementary Note 6). For practical use, we find that an estimate based on the Laplace approximation to the posterior density, combined with external information about the fluorescent background and PSF shape, performs well in a wide range of experimentally relevant conditions. The CRLB formula has a more limited range of validity, and is more sensitive to unphysical fit parameters.

Since we have limited ourselves to 2D localization using conventional optics, an extension to three dimensions (3D) is a

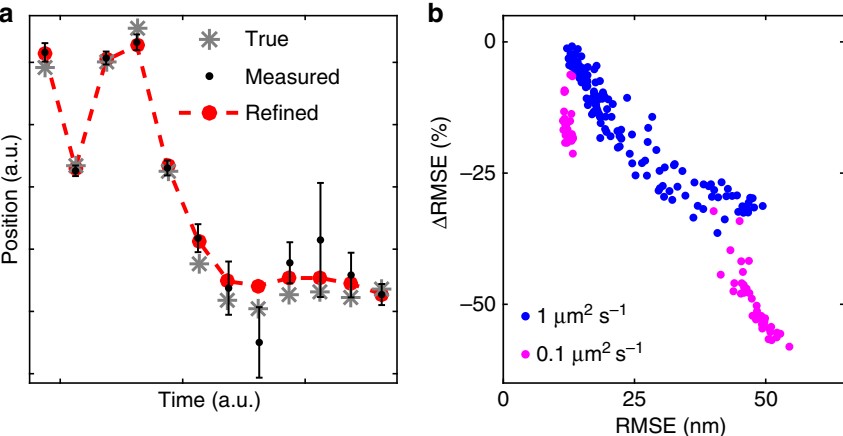

**Figure 7 | Position refinement.** Improved localization precision by modelling particle motion with the Berglund HMM. (**a**) Illustration of true, measured (± s.d.) and HMM-refined positions. (**b**) Relative change of RMSE HMM refinement, for every frame in Fig. 6a, coloured according to the true hidden state.

natural next step. We expect that useful Laplace estimators can be designed also for 3D localization precision, and that our strategy to combine realistic simulations, numerically informed para-meterizations of the localization models and physics-based priors will be helpful in achieving this. Since SMeagol accepts user-defined PSF models[11], extending it to simulate 3D localization based on engineered PSFs[29,30] or dual-plane imaging[31] is mainly a question of implementing the appropriate PSF models with enough accuracy, and should present no major difficulty. Three-dimension localization techniques are inherently asymmetric and yield different in-plane and axial precision[5], which further underscores the need for downstream analysis methods to incorporate heterogeneous localization uncertainty.

Most super-resolution microscopy applications are however not aimed at particle tracking, but imaging. For PALM/STORM-type imaging of fixed samples, our MAP localization methods does improve the precision somewhat, but the precision estimates may also be used to optimize the resolution of the final image. A simple possibility is to use the precision estimates to omit the most uncertain points from the analysis, although this may set up a difficult trade-off between localization errors and sampling density. A more efficient approach that avoids this trade-off may be to use the precision estimates as an additional input to downstream structural analysis[32]. The modelling approach may also have interesting consequences for live cell imaging, since the same fluorophore may be detected in different positions over different frames if the target is moving. For this case, we show in Fig. 7 that the combination of estimating uncertainty and modelling the fluorophore motion can produce refined position estimates, in principle pushing the localization errors below the single-image CRLB, by merging information from consecutive frames in an optimal way.

## Methods

**Synthetic data.** We generated synthetic microscopy data using SMeagol, a soft-ware for accurate simulations of dynamic fluorescence microscopy at the single molecule level[11]. We modelled the optics using the PSFgenerator[20] implementation of the Richards–Wolf PSF model[19], with 639 nm wavelength and NA = 1.4. This is a circularly symmetric PSF, appropriate for isotropic point sources or fluorophores with high rotational mobility[21]. For the EMCCD camera, we use 80 nm pixels, model EM register noise using the high-gain approximation[4,17] with EM gain 50, and add Gaussian readout noise with s.d. 10.

For localization and diffusion estimation tests, we simulated simple diffusion ($D = 1\,\mu m^2\,s^{-1}$) in a cylinder of length 20 μm and diameter 0.8 μm, similar to long *E. coli* cells, to avoid confinement artifacts in the longitudinal direction. We generated multiple data sets of 10,000–30,000 points, spanning a wide range of conditions by combining different values of exposure time (1, 3, 6, 10, 16 or 22 ms), background fluorescence (1 or 3 photons per pixel) and average spot brightness

(100–600 photons per spot). For estimating diffusion constants, we combined time steps of 3, 10 and 30 ms with various exposure times.

For the simulated multi-state data, we hand-modified a single SMeagol input trajectory from a simulated two-state model to contain four binding events with different durations and *z* coordinates as seen Fig. 6a, and also thinned out the input trajectory to create more variability in the particle paths between different realizations. We then simulated many realizations from this input trajectory, using the same PSF and camera noise as above, continuous illumination with a sample time of 5 ms, an average spot intensity of 300 photons per spot, and a time-dependent background that decays exponentially from 0.95 to 0.75 background photons per pixel with a time-constant of 0.75 s.

**Real data.** For estimating localization errors in the real imaging conditions, we use immobilized fluorescent beads with the diameter of 0.1 μm (TetraSpeck Fluorescent Microspheres, ThermoFischer T7284). The beads where diluted in ethanol and then placed on a coverslip where we let them dry in before adding water as a mounting medium.

Imaging was done with a Nikon Ti-E microscope, which was configured for EPI-illumination with a 514 nm excitation laser (Coherent Genesis MX STM) together with matching filters (Semrock dichroic mirror Di02-R514 with emission filter Chroma HQ545/50M-2P 70351). Intensity modulation was made possible by an acousto-optic tunable filter (AOTF) (AA Opto Electronics, AOTFnC) that was triggered by a waveform generator (Tektronix, AFG3021B). The waveform used was a sequence of square pulses, high for 200 and low for 1,800 ms. The two illumination intensities, high and low, corresponds to 10.7 and 0.63 kW cm$^{-2}$, respectively.

Fluorescent beads where viewed through a × 100 (CFI Apo TIRF × 100 oil, NA = 1.49) objective with a × 2 (Diagnostic instruments DD20NLT) extension in front an Andor Ultra 897 EMCCD camera (Andor Technology). This configuration puts the pixel size to 80 nm, which is the same pixel size set in the simulated data. The data set constituted of 1,000 frames (Fig. 4a) with an exposure time of 30 ms. EMCCD noise characteristics (gain, offset, readout noise) were determined by analysing a dark movie obtained with the shutter closed.

**Localization.** We perform MLE localization using an EMCCD noise model that include EM register and readout noise[4,17], which relates the probability $q(c_i|E_i)$ of the offset-subtracted pixel count $c_i$ for a given pixel intensity $E_i$ (expected number of photons per frame) in pixel $i$. For the intensity $E(x, y)$, we model the spot shape with a symmetric Gaussian,

$$E(x,y) = \frac{b}{a^2} + \frac{N}{2\pi\sigma^2}\exp\left(-\frac{(x-\mu_x)^2 + (y-\mu_y)^2}{2\sigma^2}\right), \quad (2)$$

with pixel size $a$, background $b$ (expected number of photons per pixel), spot width $\sigma$, amplitude $N$ (expected number of photons per spot) and spot position ($\mu_x, \mu_y$), and approximate the pixel intensity

$$E_i = \int_{\text{pixel } i} E(x,y)\,dx\,dy \quad (3)$$

by numerical quadrature[33]. For localization with an asymmetric Gaussian, we instead modelled the spot intensity by two principal widths $\sigma_{1,2}$ and a rotation angle (Supplementary Note 7). The log likelihood of an image containing a single

spot is then given by

$$\ln L(\theta) = \ln q_0(\theta) + \sum_{i \in \text{ROI}} \ln q(c_i | E_i(\theta)), \qquad (4)$$

where $\theta = (\mu_x, \mu_y, b, N, \sigma)$ are fit parameters, $q_0$ is a prior distribution (we set $\ln q_0 = 0$ for MLE localization), and we use a $9 \times 9$ pixel region of interest (ROI). To avoid the complications of spot identification, we use known positions to determine the ROI and initial guess for $(\mu_x, \mu_y)$.

As detailed in Supplementary Note 8, we only retain fits that converge, end up at most 4 pixels away from the true position (that is, mostly inside the ROI), yield estimated uncertainties smaller than 16 pixels, and give a spot width smaller than 9 pixels (the ROI width). For conditional RMSE boxplots, we use 7.5 nm bins and include imaging conditions with at least 300 spots per bin.

**CRLB.** The CRLB is a lower bound on the variance of an unbiased estimator[34]. We use an accurate approximation to the CRLB for a symmetric Gaussian spot from ref. 5,

$$\varepsilon_{\text{CRLB}}^2 = 2 \frac{\sigma_a^2}{N} \left( 1 + 4\tau + \sqrt{\frac{2\tau}{1 + 4\tau}} \right), \qquad (5)$$

with $\tau = 2\pi \sigma_a^2 b/(Na^2)$, $\sigma_a^2 = \sigma^2 + a^2/12$. The prefactor 2 accounts for EMCCD excess noise[4].

**Laplace approximation.** An alternative way to approximate the uncertainty of the fit parameters is to Taylor expand the likelihood around the maximum-likelihood parameters $\theta^\star$ to second order, that is

$$\ln L(\theta) \approx \ln L(\theta^*) + \frac{\partial \ln L}{\partial \theta}|_{\theta^*} (\theta - \theta^*) \\ + \frac{1}{2}(\theta - \theta^*)^T \frac{\partial^2 \ln L}{\partial \theta^2}|_{\theta^*}(\theta - \theta^*). \qquad (6)$$

The first-order term is 0, since $\theta^\star$ is a local maximum. This approximates the likelihood by a Gaussian with covariance matrix given by the inverse Hessian, that is, $\Sigma = [\partial^2 \ln L/\partial \theta^2]^{-1}$. In a Bayesian setting, this expresses the (approximate) posterior uncertainty about the fit parameters. The estimated uncertainties (posterior variances) are given by the diagonal entries of the covariance matrix, for example, $\varepsilon_{\text{Lap.}}(\mu_x) = \Sigma_{\mu_x, \mu_x}$. We compute the Hessian numerically using Matlab's built-in optimization routines, and use the log of the scale parameters $b$, $N$, $\sigma$ for fitting, since they are likely to have a more Gaussian-like posterior[35].

**Prior distributions.** For MAP localizations in the main text, we used weak normal priors with standard deviation $\ln(30)$ centred on the true value for the log background intensity. For the spot width, we define $\sigma = \sigma_0(1 + \Delta\sigma)$, where $\sigma_0 = 0.21\lambda/\text{NA}$ is the width of a focused spot, and put exponential priors with mean value one on $\Delta\sigma$. For asymmetric Gaussian spots with two principal widths, we define analogous excess widths $\Delta\sigma_{1,2}$ and use independent exponential priors with mean value 1/2. Other parameters were left with flat priors (for details, see Supplementary Note 3).

**Covariance-based diffusion estimator.** If $x_k$ ($k = 0, 1, \dots$) is the measured trajectory of a freely diffusing particle with diffusion constant $D$, the widely used model for camera-based tracking by Berglund[28] predicts that the measured step lengths $\Delta x_k = x_{k+1} - x_k$ are zero-mean Gaussian variables with covariances given by

$$\langle \Delta x_k^2 \rangle = 2D\Delta t(1 - 2R) + 2\varepsilon^2, \quad \langle \Delta x_k \Delta x_{k\pm 1} \rangle = 2D\Delta t R - \varepsilon^2, \qquad (7)$$

and uncorrelated otherwise. Here, $0 \le R \le 1/4$ is a blur coefficient that depends on how the images are acquired (for example, $R = 1/6$ for continuous illumination), $\Delta t$ is the measurement time-step, and $\varepsilon^2$ is the variance of the localization errors. Substituting sample averages for $\langle \Delta x_k^2 \rangle$ and $\langle \Delta x_k \Delta x_{k\pm 1} \rangle$ and solving for $D$ yields a covariance-based estimator with good performance[9]. If $\varepsilon^2$ is known or can be estimated independently, the first relation in equation (7) alone yields a further improved estimate of $D$. As we argue in Supplementary Note 9, these estimators apply also for variable localization errors if $\varepsilon^2$ is replaced by the average $\langle \varepsilon^2 \rangle$.

**Maximum likelihood and multi-state diffusion.** The Berglund model[28] can also be used directly for maximum-likelihood inference, which allows pointwise errors to be modelled[10]. The basic assumption is to model the observed positions $x_k$ as averages of the true diffusive particle path $y(t)$ during the camera exposure, plus a Gaussian localization error, that is,

$$x_k = \int_0^{\Delta t} y(k\Delta t + t)f(t)\mathrm{d}t + \varepsilon_k \xi_k, \qquad (8)$$

where $f(t)$ is the normalized shutter function[28], $\varepsilon_k$ is the localization uncertainty (s.d.) at time $k$, and $\xi_k$ are independent $N(0, 1)$ random numbers. Continuous illumination is described by a constant shutter function, $f(t) = 1/\Delta t$. The opposite limit of instantaneous measurement (no blur) is described by a delta function for $f(t)$, which reduces equation (8) to a standard Kalman filter[12], and leads to $R = 0$ in equation (7).

In Supplementary Note 10, we derive a maximum-likelihood estimator that learns both $D$ and $y(t)$. In Supplementary Note 11, we extend the model to multi-state diffusion, by letting the diffusion constant switch randomly between different values corresponding to different hidden states in an HMM, and derive a variational EM algorithm for maximum-likelihood inference of model parameters, hidden states and refined estimates of the measured positions.

To interpret estimated diffusion constants from simplified models, one may 'derive' corrected diffusion estimates $D^*$ by equating expressions for the step-length variance $\langle \Delta_k^2 \rangle$ from equation (7) with and without those effects present. For the Kalman ($R = 0$) and vbSPT ($R = \varepsilon = 0$) models, we get

$$D^* = \frac{D_{\text{Kalman}}}{1 - 2R}, \quad \text{and} \quad D^* = \frac{D_{\text{vbSPT}} - \langle \varepsilon^2 \rangle/\Delta t}{1 - 2R}, \qquad (9)$$

respectively, which is what we use in Fig. 6e.

**Data availability.** The data that support the findings of this study is available from the corresponding author upon reasonable request. Documented Matlab code for localization and EM-HMM analysis is freely available at https://github.com/bmelinden/uncertainSPT. Scripted examples for generating synthetic data are appended in Supplementary Data 1.

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

## Acknowledgements

We thank David Fange and Irmeli Barkefors for their careful reading of the manuscript. This work was supported by the European Research Council (Grant No. 616047), Vetenskapsrådet, the Knut and Alice Wallenberg Foundation, and the Swedish strategic research programme eSSENCE.

## Author contributions

M.L., V.C. and J.E. designed the project. M.L. and V.C. designed and implemented data analysis algorithms. E.A. designed and ran microscopy experiments. V.C. and M.L. analysed data. All authors wrote the paper.

## Additional information

**Competing interests:** The authors declare no competing financial interests.

**Publisher's note**: 

