## [Peer Review File · Nature Communications]

Reviewers' comments:

Reviewer #1 (Remarks to the Author):

In the manuscript "Pointwise error estimates in super-resolution microscopy" the authors investigate the performance of different methods for the analysis of image data intended for single particle tracking. In particular, by utilizing real and synthetic data the authors test the performance of various methods with respect to the estimated localization error. Additionally, the authors propose modification to the tested methods that lead to improved performance. An efficient solution to the localization problem and its validation are important steps towards the full comprehension of image data, especially of data from live imaging. To date no uniformly accepted solution has been proposed. Instead a number of different methods with varying rigorousness are in use. It is very pleasing that the authors of the manuscript focus into this subject.

The paper should be published.

It is the reviewer's belief that, prior to publication, the manuscript can be improved by considering the following comments:

1) The MAP estimates of this study seem to rely rather heavily on the choice of the prior distributions in at least two ways: (i) prior distributions have a considerable affect on posteriors which rises the question of how posteriors are affected by the uncertainty in the priors that might require extensive calibration and (ii) the numerical solution of the localization problem as explained in the subsequent comments.

2) The solution of the localization problem through MLE or MAP is followed by a classification step where diverging fits are discarded. It is the reviewer's experience that such classification introduces artifacts as it preferably excludes fits that otherwise would contribute to the estimated locations.

3) Clearly, the probability landscape is affected by the particular choices of the priors which have extensive overlap with the posteriors. It is likely that the probability landscape leads to divergence and subsequently rejection of certain pixel populations.

4) Similar concerns arise from the choice of the starting points for the optimizations on the MLE or MAP. Generally, the optimizer might be trapped in local maxima and the performance of the methods needs to be evaluated taking in account this possibility. By starting the optimizers essentially at the optimal point it is ensured that the optimizer converges to the global maximum. However, this is possible only for the artificial datasets and not for the experimental ones.

Reviewer #2 (Remarks to the Author):

The authors present an improved version of the estimation of the uncertainty in the context of localization microscopy. This is an relevant question to be really quantitative and for time-varying localization study. Better estimation of the uncertainty will give better reconstruction. The Cramér-Rao Bound is usually used in the field even the assumptions are not always fulfilled in practice. The authors are proposing new methods to estimate the uncertainty for the localization microscopy and they show significant improvements.

The manuscript is well written and very well-documented. Many experiments have been achieved and reported here which are convincing enough.

Main concerns:

- The manuscript will be more outstanding by moving the technical aspects in the supplementary information. The experiments can be summarized in the main text and details can be reported in the supplementary information. By this way, the attention will be attracted by the main claims of this work.
- The paper is limited to the Gaussian theoretical PSF. Do the authors draw the same conclusions with a experimental PSF, or with a better PSF (Airy pattern with several lobes)? At least this point should be commented in the discussion.
 - The Matlab code is open. In the same spirit, is it possible to open the data? in particular the synthetic data such a way that the readers can explore the effects on the different perturbations.
- This work is limited to 2D. It is primarily important to have this study in 3D, both for engineered PSF and for multiple planes acquisition.
- The authors have a strong claim at the end of the discussion: "in principle pushing the localization errors below the Cramér-Rao... by merging information from consecutive frames". This claim should be discussed, proven and illustrated by experiments.

Minor issues:

- title: more focus on localization microscopy, finally localization is only one of the super-resolution techniques in microscopy
- bibliography reference is starting at [2]
- Confusion between term: accuracy / precision / uncertainty, e.g. line 11, we should read localization precision and not localization accuracy.
- Figure 1 is illustrating only the effect of the number of photons and the background, which are obvious for everybody. More interesting is to illustrate, the photobleaching, the out-of-focus, ...
- Caption of figure: inverse $b = 1$, $N = 150$ to $N=150$, $b=1$
- Figure 2a) missing bracket before nm
- line 70: it is not clear what it is the noise model of the EMCCD camera, is it included the register gain?
- line 213, missing space

Reviewer #3 (Remarks to the Author):

Linden et al. report on a technique to improve the localization of point emitters under unfavourable circumstances, mainly due to low photon count, high and heterogeneous background, and focus errors. Under these circumstances the fitting routine can get stuck in a local optimum in fit parameter space that does not correspond to the true values of the parameters. Several measures for the localization uncertainty, in particular closed-form formulas are then no longer valid. The authors have, in my view correctly, identified this issue and propose a solution. The key innovation is to use prior knowledge on spot width and background to guide the optimizer to the "right" solution.

Although this finding is worthy of publication in some form I am hesitant to advise publication in its present form. I have several points of criticism:

1. The authors report on a simulation study, which is in no way a substitute for a real experiment. Despite the claimed sophistication of the model that is used, it will always fall short in the confrontation with real practice. In short, the burden of evidence is a bit higher.
2. The prior knowledge is encoded in log-normal and skewed log-normal distributions. It is unclear how these are motivated other than by trial-and-error (fig S2). I could envision a simulation or experimental study in which the actual distribution of fitted background or spot width is measured/computed and subsequently fitted with log-normal, skewed log-normal, etc. distributions.
3. There is clear prior information concerning spot width from λ/NA , so that could work straight

away. There is no objective information concerning background, which can moreover be heterogeneous. The authors are sketchy about how the "true" background can be obtained from real data. I think they should be more explicit and also expand the analysis. In addition, a comparison to the temporal median filter approach of Hoogendoorn et al. (Scientific Reports 4, 3854, 2014) is in order.

4. An alternative to the use of a priori distributions for parameters is the use of a posteriori filtering, which is now a commonly used procedure for outlier removal. Minimum and maximum values for spot width, photon count or for goodness of fit are in use. This is more ad-hoc than the proposed method but could nevertheless work equally well. The authors should compare their method to this approach.

5. The authors claim that their simulation package is "highly realistic", yet their PSF is based on Gibson-Lanni, which neglects vectorial/polarization aspects of the imaging which are definitely relevant at high NA. A disclaimer is a must here.

6. With their suppression of outlier fits I would expect that localization microscopy images would improve, although the authors do not seem to claim that. It is worth the effort to check that and perhaps even to quantify it using e.g. FRC resolution.

7. I find the nomenclature distinguishing "CRLB" from "Laplace approximation" confusing. What they call "CRLB" I would call "CRLB formula" and what they call "Laplace approximation" I would call "true CRLB" or "CRLB from fit". If the authors wish to stick to their terminology they should explain what they mean when the terms are introduced. It took me a while to figure out what they meant by "Laplace approximation" which is after all a very generic term.

Response to referee comments:

Reviewer #1 (Remarks to the Author):

In the manuscript “Pointwise error estimates in super-resolution microscopy” the authors investigate the performance of different methods for the analysis of image data intended for single particle tracking. In particular, by utilizing real and synthetic data the authors test the performance of various methods with respect to the estimated localization error. Additionally, the authors propose modification to the tested methods that lead to improved performance. An efficient solution to the localization problem and its validation are important steps towards the full comprehension of image data, especially of data from live imaging. To date no uniformly accepted solution has been proposed. Instead a number of different methods with varying rigorousness are in use. It is very pleasing that the authors of the manuscript focus into this subject.

The paper should be published. It is the reviewer’s belief that, prior to publication, the manuscript can be improved by considering the following comments:

1) The MAP estimates of this study seem to rely rather heavily on the choice of the prior distributions in at least two ways: (i) prior distributions have a considerable affect on posteriors which rises the question of how posteriors are affected by the uncertainly in the priors that might require extensive calibration and (ii) the numerical solution of the localization problem as explained in the subsequent comments.

Reply: We thank the referee for raising these very relevant points, that we believe are more clearly described in the revised manuscript.

Regarding (i), our revised analysis uses more inclusive criteria for which dots to include in the analysis, which give a less biased and more nuanced comparison between true and estimated precision. As a result, the maximum-likelihood methods perform better, and thus weaker priors are needed, which means that uncertainty in the priors is less of a concern. We also formulated new functional forms for the priors so that the prior parameters are easier to interpret, and performed sensitivity tests.

In particular,

- The prior on the fluorescence background is weaker than in the previous version, and no accurate calibration is needed. This is detailed in SI S5, and more briefly in the results section on the maximum a posteriori method.
- There is an explicit minimum spot width parameter, which depends on experimental parameters that are easily accessible, as discussed on lines 160-175.
- Finally, there are prior parameters describing the variation of the PSF width relative to the minimum width parameter. This distribution reflects complex relationships between the imaging system, experimental conditions, and sample geometry (SI Sec S4) , and cannot be set with absolute certainty. Instead, we ran numerical tests of the robustness against variations in these parameters (detailed in SI Sec S5, figures S15 and S16). We found that the symmetric spot model is quite robust against variations in this parameter. The asymmetric spot model is less robust, but can perform better if the prior is properly calibrated (Fig 2-3, SI Sec. 5.3-5.4).
- We tested the sensitivity to calibration errors in the EM gain, and found a weak dependence for the estimated precision, and a scaling argument that seems to describe it (SI Sec. S5.6).

Point (ii) is discussed below.

2) The solution of the localization problem through MLE or MAP is followed by a classification step where diverging fits are discarded. It is the reviewer's experience that such classification introduces artifacts as it preferably excludes fits that otherwise would contribute to the estimated locations.

Reply: we thank the reviewer for this comment. As outlined above, and detailed in SI Sec. S2, we re-examined our threshold for discarding spots, and modified them to be more inclusive in order to better capture the true averages over the whole dot shape distribution. We also came up with the conditional error plot as a complementary way to make this comparison. Finally, we also experimented with using a dot detection algorithm (SI Sec. 5.7), to verify that our conclusions remain valid for dot ensembles mimicking those in real experiments.

With these modifications, we believe that our conclusions are more robustly established.

3) Clearly, the probability landscape is affected by the particular choices of the priors which have extensive overlap with the posteriors. It is likely that the probability landscape leads to divergence and subsequently rejection of certain pixel populations.

Reply: it seems the referee is curious about the rejection rates in the different cases and concerned that the performance we see do not reflect the performance one should expect when analyzing real data with no ground truth.

First, we agree that the fraction of discarded spots is a relevant quantity to include, and have added this information in the SI, Sec S3.2 and S5.5. We believe that the most important observation here is that the overall convergence rate is high, except for the MLE fit with the asymmetric spot model (which we do not recommend using, and do not mention in the main text).

Second, in addition to improving the consistency of the precision estimates, the priors also enable a higher fraction of localizations to converge properly and to decreased localization errors (SI Sec. S5.5).

It is not clear to us that the "extensive overlap" between priors and posteriors is a cause for concern. However, we note that the revised priors are considerably wider than the posteriors, with the exception that PSF widths below the diffraction limit are excluded. This has a clear physical motivation however, and is also corroborated by the high intensity simulations described in Sec. S4.

4) Similar concerns arise from the choice of the starting points for the optimizations on the MLE or MAP. Generally, the optimizer might be trapped in local maxima and the performance of the methods needs to be evaluated taking in account this possibility. By starting the optimizers essentially at the optimal point it is ensured that the optimizer converges to the global maximum. However, this is possible only for the artificial datasets and not for the experimental ones.

Reply: This is a valid concern. Similarly, we select our regions of interest based on the known spot positions, which means that we include all possible spots in our statistics, including those that would be practically undiscoverable due to strong defocus or extreme fluorophore motion during exposure. We also make sure that we only fit single spots and do not need to consider the complications associated with multiple spots in close vicinity.

We believe that these kinds of idealizations are useful in the sense that they enable us to separate the problem of estimating precision from the complications of spot finding, multi-spot fitting, and global optimization, and focus on the main statistical question, estimating localization precision.

However, to complement this perspective, we added numerical experiments where we used a dot detection algorithm to filter out hard-to-detect spots and supply less ideal initial conditions and regions of interest. The results (see SI sec. S5.7, also referred to in the discussion) differ quantitatively from those of the main text, mainly because spots that are difficult to detect also tend to represent ill-posed localization problems, and thus the dot population returned by the dot detection algorithm is overall more well-behaved. We do not see signs of worse optimization performance due to the less ideal initial conditions.

Reviewer #2 (Remarks to the Author):

The authors present an improved version of the estimation of the uncertainty in the context of

localization microscopy. This is an relevant question to be really quantitative and for time-varying localization study. Better estimation of the uncertainty will give better reconstruction.

The Cramér-Rao Bound is usually used in the field even the assumptions are not always fulfilled in practice. The authors are proposing new methods to estimate the uncertainty for the localization microscopy and they show significant improvements.

The manuscript is well written and very well-documented. Many experiments have been achieved and reported here which are convincing enough.

Main concerns:

- The manuscript will be more outstanding by moving the technical aspects in the supplementary information. The experiments can be summarized in the main text and details can be reported in the supplementary information. By this way, the attention will be attracted by the main claims of this work.

Reply: We have moved as much as possible of the technical material to the methods or SI. We believe that the parts that remains are needed for correctness, and also note that the SI is already quite long. However, we would of course be grateful for suggestions for additional things to move.

- The paper is limited to the Gaussian theoretical PSF. Do the authors draw the same conclusions with a experimental PSF, or with a better PSF (Airy pattern with several lobes)? At least this point should be commented in the discussion.

Reply: First, we would like to note that expect SI sec. S6 (see below), we use Gaussian PSF models only for localizations, while the simulated data described in the main text is generated with a much more realistic PSF. As detailed in reply to Q4 of referee 3, we have also switched PSF model for the revision, and reached essentially the same conclusions.

That said, the referee raises an important point, namely that the Gaussian spot models are not completely accurate models for the intensity profiles of actual spots. One might reasonably suspect that more realistic models might do better. One possibility is to measure and parameterize the PSF of ones microscope, for example as described by Liu et al (our ref 25).

To address this question as simple as possible, we generated simulated data using a (z-dependent) Gaussian PSF, and analyzed it with our standard MLE localization methods using Gaussian spot models. Although this in one sense represents perfect agreement between model and data-generating PSF, the results (SI sec. S6, and discussed around line 200) show only a limited improvement, and the bias artifacts seen with the more realistic PSF remains.

Since the Gaussian PSFs used for this simulation still had a z-dependence, we believe the reason may be that real spots under these conditions are mixtures of different PSFs, since they contain photons emitted at different positions and degree of defocus. Thus, perfect agreement between the fit model and the PSF of the microscope does not mean that the fit model is a perfect model for the actual dots.

- The Matlab code is open. In the same spirit, is it possible to open the data? in particular the synthetic data such a way that the readers can explore the effects on the different perturbations.

Reply: We agree that it would be a good thing for users of these methods to experiment themselves. However, our numerical experiments are organized so that synthetic data is generated, analyzed and then discarded, and only the results are stored. Instead, we have added examples scripts as supplementary data, which together with the uncertainSPT and SMeagol packages can be used to generate synthetic data as in our numerical experiments.

- This work is limited to 2D. It is primarily important to have this study in 3D, both for engineered PSF and for multiple planes acquisition.

Reply: We agree that point-wise precision estimates of localization precision would be very useful in 3D. However, establishing localization errors in the 2D case is already a significant advance, and as there are several fundamentally different methods for 3D imaging and tracking currently in use (e.g., biplane, astigmatism, double helix, iPALM), an extension to 3D comes with a large parameter space to cover. We do not believe that such an extension would be realistic to undertake as part of the present study. However, we do believe that our development of the 2D case have yielded both useful methodologies and practical lessons that are highly relevant to the 3D case.

SMeagol is capable of simulating 3D problems. Mainly, one needs to develop and implement new and realistic PSF models. Astigmatic, double helical, or other single-image PSFs are easily added to SMeagol simulations, as described in the software documentation. We believe that multiple plane acquisition would also be doable within the current SMeagol framework, by putting the different planes side by side (but this is not covered in the documentation). The method of high-intensity simulations should also be directly applicable to the 3D case.

For the inference models, our experience from developing the 2D case is that the parameterization of the localization models are important, to make the Laplace approximation as well as the optimization problem work well. This step also requires physical insight however, and our impression is that the 3D techniques are less well understood than 2D, so that some exploratory work may be needed. For example, we are not aware of any systematic studies on the effects of motion blur on 3D localization.

We have modified our discussion of 3D localization (around line 380) to outline our views on extension to 3D more clearly.

- The authors have a strong claim at the end of the discussion: "in principle pushing the localization errors below the Cramér-Rao... by merging information from consecutive frames". This claim should be discussed, proven and illustrated by experiments.

Reply: The claim we would like to make is simply that the CRLB is context dependent, and can be improved upon by using additional information. We believe this is an interesting observation in this context, but not a very strong claim that require extensive discussion or validation.

The key point is that a CRLB is a property of the type, quality, and amount of data that goes into an analysis. If any of those inputs change, the CRLB changes as well. For example, localizations based on 1000 photons are not bound by the CRLB for 100 photons, and localizations based on multiple images are not bound by the CRLB of a single image.

In this case, our HMM analysis with estimated precision uses positions from a sequence of images, and the assumption of multi-state diffusive motion, to estimate the true position behind every localization. Each of these "input" localizations are based on just a single image, and it is therefore not surprising that the HMM analysis can do better. Also note that this argument does not require that the single image analysis is optimal and reaches its CRLB. The CRLB for the single-image localization problem simply does not apply to the multi-image analysis.

We have modified the formulation to emphasize the context-dependence and stress that the comparison is with the CRLB of single images: "... in principle pushing the localization errors below the **single-image** Cramer Rao lower bound, by merging information from consecutive frames in an optimal way." A similar formulation appears early in the "position refinement" section: " ..., and in principle beat the Cramer-Rao lower bound for **single image** localizations." (Our bold face).

Minor issues:

- title: more focus on localization microscopy, finally localization is only one of the super-resolution techniques in microscopy

Reply: we changed the title to "Pointwise error estimates in localization microscopy", and introduced similar changes in various places throughout the main text.

- bibliography reference is starting at [2]

Reply: This has been fixed in the revised version.

- Confusion between term: accuracy / precision / uncertainty, e.g. line 11, we should read localization precision and not localization accuracy.

Reply: we thank the referee for this suggestion, and have switched to using localization precision as our main term in the revised version.

- Figure 1 is illustrating only the effect of the number of photons and the background, which are obvious for everybody. More interesting is to illustrate, the photobleaching, the out-of-focus, ...

Reply: Good point. We have remade figure 1 to include systematic variations of spot intensity, out-of-focus, and motion blur.

- Caption of figure: inverse $b = 1$, $N = 150$ to $N=150$, $b=1$

- Figure 2a) missing bracket before nm

Reply: Good points, fixed in the revised version.

- line 70: it is not clear what it is the noise model of the EMCCD camera, is it included the register gain?

Reply: Yes. The noise model we use is detailed in for example the supporting information of Mortensen et al (2) , section 4, "What we *really* see with an EMCCD", which is cited just after that sentence. We also added some wording to further clarify that the EMCCD noise in the methods section.

- line 213, missing space

ML: Thank you. Fixed.

Reviewer #3 (Remarks to the Author):

Linden et al. report on a technique to improve the localization of point emitters under unfavourable circumstances, mainly due to low photon count, high and heterogeneous background, and focus errors. Under these circumstances the fitting routine can get stuck in a local optimum in fit parameter space that does not correspond to the true values of the parameters. Several measures for the localization uncertainty, in particular closed-form formulas are then no longer valid. The authors have, in my view correctly, identified this issue and propose a solution. The key innovation is to use prior knowledge on spot width and background to guide the optimizer to the “right” solution.

Although this finding is worthy of publication in some form I am hesitant to advise publication in its present form. I have several points of criticism:

1. The authors report on a simulation study, which is in no way a substitute for a real experiment. Despite the claimed sophistication of the model that is used, it will always fall short in the confrontation with real practice. In short, the burden of evidence is a bit higher.

Reply: We do agree that real live cell data can be more complicated than both simulations and *in vitro* experiments. However, it is difficult to validate analysis methods without access to some ground truth, and ground truth with respect to moving molecules in a living cell would be very difficult to come by.

(Our recent paper “Nanometer resolution imaging and tracking of fluorescent molecules with minimal photon fluxes”, *Science* Dec 2016, comes very close to ground truth, but would be impossible to combine with simultaneous camera based tracking).

The use of simulated data not only offers exact ground truth, but also the possibility to control and separate different types of difficulties. This makes it possible to study the main statistical questions carefully without uncontrolled confounding factors. In short, we believe that careful simulations are the best approach in this particular case.

By using freely available simulation software and releasing scripts that reproduce key numerical experiments, we also make it possible for interested readers to run numerical experiments tuned to their own particular needs. We believe that this will contribute to a general increased level of rigor for data analysis in the field.

2. The prior knowledge is encoded in log-normal and skewed log-normal distributions. It is unclear how these are motivated other than by trial-and-error (fig S2). I could envision a simulation or

experimental study in which the actual distribution of fitted background or spot width is measured/computed and subsequently fitted with log-normal, skewed log-normal, etc. distributions.

Reply: see below.

3. There is clear prior information concerning spot width from λ/NA , so that could work straight away. There is no objective information concerning background, which can moreover be heterogeneous. The authors are sketchy about how the “true” background can be obtained from real data. I think they should be more explicit and also expand the analysis. In addition, a comparison to the temporal median filter approach of Hoogendoorn et al. (Scientific Reports 4, 3854, 2014) is in order.

Reply to 2 and 3: we agree that the physical and statistical considerations behind the prior choice could be improved, and have revised both the motivation and choice of priors do this:

- We have switched to a more transparent functional form for the priors, which includes an explicit lower bound on the PSF as suggested by the referee.
- We added a description of the physical and statistical reasoning behind the prior choice to the main text (lines ~170).

Regarding background estimation, the revised prior includes only a weak background dependence, which only requires an order-of-magnitude background estimation to parameterize, as shown in Fig S14, which also implies some robustness against background variations. Thus, we believe a simple inspection will suffice in many cases, and the median filter approach of Hoogendoorn et al (3) is cited in this context. Since an accurate background estimate is not needed, we do not think that an in-depth treatment of how to estimate background fluorescence is motivated.

4. An alternative to the use of a priori distributions for parameters is the use of a posteriori filtering, which is now a commonly used procedure for outlier removal. Minimum and maximum values for spot width, photon count or for goodness of fit are in use. This is more ad-hoc than the proposed method but could nevertheless work equally well. The authors should compare their method to this approach.

Reply: We are not sure that a posteriori filtering and our use of priors are always means to the same end. We use priors not to weed out false positives or bad fits, but rather to improve the precision estimates in true positives. In some applications, one may wish to use the estimated precision as an additional parameter for a posteriori filtering, and separate methods to filter out false positives may also be needed in some applications, but these methods are not the main focus of this work.

That said, we have included tests of some simple parameter filters in SI section S3.2, which is also briefly discussed in the main text (lines ~140-150). Briefly, filtering on fit parameter values can improve the consistency of the precision estimates, but the fraction of spots that must be thrown away can be very significant in low light conditions (up to 50%). This becomes problematic especially in single particle tracking applications, where trajectories are chopped into short segments. The approach using priors does not suffer from such a trade-off (fig S15).

5. The authors claim that their simulation package is “highly realistic”, yet their PSF is based on Gibson-Lanni, which neglects vectorial/polarization aspects of the imaging which are definitely relevant at high NA. A disclaimer is a must here.

Reply: We thank the referee for pointing this out.

The SMeagol software is highly flexible with regards to PSF models, and we have rerun all numerical experiments using another PSF model by Richards & Wolf (4), which does include vectorial aspects. The images below show a comparison of the R&W and G&L models, which indeed show some differences. For example, the R&W model has a smaller z-range of well-behaved spots and less pronounced secondary rings.

However, we believe that the most important points of our simulations are that the simulated PSF is significantly different (and more realistic) compared to the PSF model used for fitting, and that motion blur and defocus effects are present in reasonable amounts to contribute random distortions of the spot shapes. This is true both for both the above PSF models, and our overall conclusions remains mainly unchanged.

Polarization effects are not included in the present version, since it would have to include also simulation of the rotational diffusion of the simulated molecules and specific aspects of labeling chemistry, and thus becomes very systems dependent. We do however acknowledge that such effects may be important for accurate localization if the dyes do not rotate freely. We have inserted short clarifications about this limitation in the main text (line 111) and in the methods section.

6. With their suppression of outlier fits I would expect that localization microscopy images would improve, although the authors do not seem to claim that. It is worth the effort to check that and perhaps even to quantify it using e.g. FRC resolution.

Reply: Indeed, one could imagine filtering the localizations according to their estimated precision in such a way that the final resolution improves. There is a trade-off to be made here though: such a strategy means that the average localization error is improved at the cost of decreased labeling density, and the overall image resolution depends on both these factors, as well as the spatial structure of the object (5). At low densities it may even be good to keep also inaccurate points.

One may also question such strategies on the grounds that they are inherently wasteful, since even uncertain localizations contain some information. A more fruitful strategy may be to try using all localizations in a way that accounts for differences in localization uncertainty, in analogy with the time series analysis behind figures 6 and 7.

We believe that it would be difficult to be specific on this point in a useful way without either a more thorough theoretical investigation, or a careful specification of some particular experimental conditions. Since we already have one detailed application of improved data analysis with precision estimates (fig 6), we prefer to limit the discussion on this topic to a general remark in the Discussion section (near line 392).

7. I find the nomenclature distinguishing “CRLB” from “Laplace approximation” confusing. What they call “CRLB” I would call “CRLB formula” and what they call “Laplace approximation” I would call “true CRLB” or “CRLB from fit”. If the authors wish to stick to their terminology they should explain what they mean when the terms are introduced. It took me a while to figure out what they meant by “Laplace approximation” which is after all a very generic term.

Reply: We agree that the CRLB that we refer to is the specific approximation given by the analytical formula in (1). This is now more clearly explained. We are reluctant to use long names in figures, but have partly followed the referees suggestion to refer to the CRLB estimator as a formula in several places in the main text.

The Laplace approximation is indeed a rather generic term, and it is perhaps unfortunate to name this estimator after a numerical approximation scheme rather than the underlying statistical concept. However, the name is firmly established in Bayesian statistics as a method for approximating the posterior density, and thus we prefer to keep it rather than inventing something new. However, we have added some language to stress the connection to the posterior density when the Laplace estimator is introduced on p 2, and also modified the abstract to mention posterior density.

References

1. Rieger, B., and S. Stallinga. 2014. The Lateral and Axial Localization Uncertainty in Super-Resolution Light Microscopy. *ChemPhysChem*. 15: 664–670.
2. Mortensen, K.I., L.S. Churchman, J.A. Spudich, and H. Flyvbjerg. 2010. Optimized localization analysis for single-molecule tracking and super-resolution microscopy. *Nat. Methods*. 7: 377–381.

3. Hoogendoorn, E., K.C. Crosby, D. Leyton-Puig, R.M.P. Breedijk, K. Jalink, T.W.J. Gadella, and M. Postma. 2014. The fidelity of stochastic single-molecule super-resolution reconstructions critically depends upon robust background estimation. *Sci. Rep.* 4.
4. Richards, B., and E. Wolf. 1959. Electromagnetic Diffraction in Optical Systems. II. Structure of the Image Field in an Aplanatic System. *Proc. R. Soc. Lond. Math. Phys. Eng. Sci.* 253: 358–379.
5. Mukamel, E.A., and M.J. Schnitzer. 2012. Unified Resolution Bounds for Conventional and Stochastic Localization Fluorescence Microscopy. *Phys. Rev. Lett.* 109: 168102.
6. Broeken, J., H. Johnson, D.S. Lidke, S. Liu, R.P.J. Nieuwenhuizen, S. Stallinga, K.A. Lidke, and B. Rieger. 2015. Resolution improvement by 3D particle averaging in localization microscopy. *Methods Appl. Fluoresc.* 3: 014003.

REVIEWERS' COMMENTS:

Reviewer #1 (Remarks to the Author):

The authors have responded to this referee's comments adequately. The only final remark I have is a recommendation for the authors: it would be nice to have in point form the assumptions made by the method such as the assumptions of Gaussian PSFs amongst many others.

Reviewer #2 (Remarks to the Author):

The authors have provide many relevant explanations in their rebuttal. The revised manuscript is improved a lot and finally it is simplified the reading. They have addressed all the points and the questions of the reviewers. So, I will fully recommend to publish this paper.

Reviewer #3 (Remarks to the Author):

The authors have done a good job improving their work based on the many comments by me and the other reviewers. I can support publication of the manuscript now.

One small remark about my point 6. The authors are correct in pointing out that throwing away too many "incorrect" fits is wasteful and reduces the labeling density in the final SR images. That was precisely why I raised this point! I think they can do better than the state-of-the-art outlier removal procedures, a point worth mentioning.

REVIEWERS' COMMENTS:

Reviewer #1 (Remarks to the Author): The authors have responded to this referee's comments adequately. The only final remark I have is a recommendation for the authors: it would be nice to have in point form the assumptions made by the method such as the assumptions of Gaussian PSFs amongst many others.

Reply: We agree with the referee that spelling out the underlying assumptions is important. However, we would also like to emphasize that not all assumptions are equally critical. A major point of this work is to find methods that are not very sensitive to deviations between model assumptions and real data, and to explore the limits of this robustness. For example, an assumed Gaussian spot intensity profile works quite well also on simulated data based on non-Gaussian PSFs, as long as motion blur effects are not too large. Similarly, the spot width prior distributions we use perform quite well even though it is not in quantitative agreement with the underlying spot shape distribution. Third, all results do not depend equally on all assumptions. We assume EMCCD camera noise in our examples, but this does not mean that for example the spot shape models trajectory analysis methods are not applicable to data acquired with other types of cameras.

We believe that a single comprehensive bullet-list of assumptions is not well suited to capture such nuances, or would become too extensive to make sense if it did. We therefore prefer to keep the present format, where the different assumptions are outlined and discussed throughout the main text, Methods section and to some extent in the Supplementary Notes.

Reviewer #2 (Remarks to the Author): The authors have provide many relevant explanations in their rebuttal. The revised manuscript is improved a lot and finally it is simplified the reading. They have addressed all the points and the questions of the reviewers. So, I will fully recommend to publish this paper.

Reviewer #3 (Remarks to the Author): The authors have done a good job improving their work based on the many comments by me and the other reviewers. I can support publication of the manuscript now.

One small remark about my point 6. The authors are correct in pointing out that throwing away too many "incorrect" fits is wasteful and reduces the labeling density in the final SR images. That was precisely why I raised this point! I think they can do better than the state-of-the-art outlier removal procedures, a point worth mentioning.

Reply: We have expanded our discussion of PALM/STORM imaging on this point to explicitly include the trade-off between localization errors and labeling density, and the possibility to avoid it by using the estimated uncertainty explicitly in the image construction.